analytical chemistry/green chemistry/ spectroscopy

etoposide, moxifloxacin, nalbuphine, synchronous fluorimetry, human urine

**Author for correspondence:**
M. M. Salim
e-mail: mmasalim@mans.edu.eg

This article has been edited by the Royal Society of Chemistry, including the commissioning, peer review process and editorial aspects up to the point of acceptance.

# Eco-friendly fluorimetric approaches for the simultaneous estimation of the co-administered ternary mixture: etoposide, moxifloxacin and nalbuphine

M. M. Tolba[1], F. Belal[1] and M. M. Salim[1,2]

[1]Department of Pharmaceutical Analytical Chemistry, Faculty of Pharmacy, Mansoura University, Mansoura 35516, Egypt
[2]Department of Pharmaceutical Chemistry, Faculty of Pharmacy, Horus University—Egypt, New Damietta 34511, Egypt

MMT, 0000-0001-7968-6487; FB, 0000-0001-6242-2178; MMS, 0000-0003-4429-6504

Antineoplastic drugs, etoposide (ETO), are widely used in leukaemia. A patient with leukaemia has a relative infection with pneumonia treated by fluoroquinolones as moxifloxacin HCL (MOX). Because opioid analgesic as nalbuphine HCL (NAL) does not have a ceiling dose, it is used to manage the distasteful sensory in leukaemia. Consequently, green methods for synchronous spectrofluorimetric quantification of a ternary mixture of ETO, MOX and NAL were developed. The *first approach* relies simply on the estimation of MOX at 371 nm by conventional synchronous fluorimetric technique ($\Delta\lambda$ of 60 nm). The *second approach* depends on applying the first derivative synchronous fluorimetric technique ($\Delta\lambda$ of 60 nm) for simultaneous estimation of ETO and NAL at 257 and 273 nm, respectively. A good linear correlation was obtained in the ranges of 0.04–0.40, 0.10–1.00 and 0.50–5.00 µg ml$^{-1}$ for MOX, ETO and NAL, respectively. Moreover, the proposed approaches were successfully applied for the estimation of the studied drugs in the pharmaceutical dosage forms. Additionally, the synchronous assessment of ETO, MOX and NAL in the spiked human urine was successfully attained by the facile protein precipitation technique. The mean % recoveries in spiked human urine were 99.49, 98.07 and 98.48 for MOX, ETO and NAL, respectively.

# 1. Introduction

Etoposide (ETO, figure 1); (5S, 5aR, 8aS, 9R)-9-(4,6-O-ethylidene-β-D-glucopyranosyloxy)-5, 8, 8a, 9-tetra hydro -5-(4-hydroxy-3,5-dimethoxy phenyl)-isobenzofuro[5,6f] is a vastly used antineoplastic drug mainly for the treatment of leukaemia and also for lymphoma through changing the configuration of DNA by complexation action [1,3]benzodioxol-6(5aH)-one [1–3]. Pneumonia assessment by bacteria is common among leukaemia patients in the hospital, and it is treated by newer fluoroquinolones as moxifloxacin HCL (MOX, figure 1) [4]. MOX is named as 1-cyclopropyl-6-fluoro-1,4-dihydro-8-methoxy-7-[(4aS,7aS)-octahydro-6Hpyrrolo[3,4-b]pyridin-6-yl]-4-oxo-3-quinolinecarboxylic acid hydrochloride [1]. Opioid as nalbuphine HCL (NAL, figure 1) is approved to control cancer-associated pain and it is named 17-cyclobutylmethyl-7, 8-dihydro-14-hydroxy-17-normorphine hydrochloride; (−)-(5R, 6S, 14S)-9a-cyclobutylmethyl-4, 5-epoxymorphinan-3, 6, 14-triol hydrochloride [1,5]. ETO is official in both United States pharmacopeia and British pharmacopeia [6,7].

Several methods have been reported in the literature for the estimation of ETO, MOX and NAL in their pharmaceutical dosage form and/or biological fluids; for ETO: spectrophotometric [8], spectrofluorimetric (SF) [9], electrochemical [10,11], high-performance thin-layer chromatography (HPTLC) [12], high-performance liquid chromatography (HPLC) [13–17], gas chromatography (GC) [18] and capillary electrophoresis (CP) [19]; for MOX: spectrophotometric [20,21], SF [22], electrochemical [23], HPTLC [24], HPLC [25–27] and CP [28] and for NAL spectrophotometric [29–31], SF [31,32], flow injection analysis [33,34], HPLC [35–37] and GC [38]. These presented methods did not guide the monitoring of these three drugs in the biological matrices.

Hence, the proposed green approaches for synchronous SF quantification of a ternary mixture of ETO, MOX and NAL in their pharmaceutical dosage form and/or biological fluids were developed. The spectrofluorimetry technique is characterized by a considerable high sensitivity, monitoring an extensive range of drugs, whether in their pharmaceutical dosage forms or their biological fluids but with moderate selectivity. Since the selectivity of spectrofluorimetry is improved by applying synchronous scanning [39], using this technique provides additional cons of the proposed methods. The significance of adding the derivative process to synchronous scanning in spectrofluorimetry technique is an additional merit that enhances the selectivity and sensitivity of the proposed approaches [40].

The greenness of the explained SF technique affirmed additional advantage to this study which was assured by applying different assessment parameters, namely national environmental methods index (NEMI), green analytical procedure index (GAPI) and analytical eco-scale [41,42]. Green chemistry presents some factors as reducing solvent utilization via providing alternatives that are less dangerous and more environmentally friendly instead of consuming toxic organic solvents and improving the recycling of large-scale preparatory research techniques [43,44].

The aim of the presented green SF technique is the quantification of a ternary mixture of ETO, MOX and NAL quickly and successfully in their therapeutic dosage formulations without interference from excipients without any pretreatment. Furthermore, this green SF technique was performed for the quantification of a mixture of these drugs in biological fluid. Additionally, the performance of the green SF technique was affirmed by applying International Council on Harmonisation (ICH) rules [45].

# 2. Experimental procedure

## 2.1. Apparatus

Synchronous fluorescence spectra were measured by a Cary Eclipse fluorescence spectrophotometer (product of Agilent Technologies, USA) which was adjusted at 800 V (high-voltage mode). Experimental study was performed using quartz cell (1 cm) and the smoothing factor was adjusted to be 19.0. pH was measured and regulated by consort NV P-901 pH meter from Belgium. A water bath (Cambridge Ltd, UK) was used for shaking purposes. The application of the proposed SF in the biological fluid was performed using different apparatus, including Vortex IVM-300 p of Gemmy Industrial Corp. (Taiwan) for mixing the urine and centrifugation using centrifuge model 2-16P (Germany).

## 2.2. Materials and reagents

— ETO, with a purity of 99.86%, was purchased from Cayman Chemicals (USA). MOX, with a purity of 99.64%, was obtained from Future Pharmaceutical Industries (Cairo, Egypt). NAL with purity 100.11% was obtained from Amoun Pharmaceutical Co., Cairo, Egypt.

**Figure 1.** The structural formulae for ETO, NAL and MOX.

— Pharmaceutical formulations of ETO, MOX and NAL were obtained from community pharmacy; including Etoposide Mylan® solution, batch#1077, tagged to include 20 mg ETO ml$^{-1}$, the product of Mylan Pharmaceuticals ULC, Etobicoke, Ontario, Canada; Moxavidex® 400 mg tablets, batch#180892A, comprise 436.8 mg MOX/tablet, the product of Future Pharmaceutical Industries, Badr City, Cairo, Egypt; and Nalufine® ampoule (20 mg NAL/ ampoule), batch#195828A, manufactured by Amoun Pharmaceutical Co., Egypt.

— The biological fluid (human urine) accumulated from a healthy female approximately 37 years old and was retained at −5°C until involved in this study.

— The surfactants involving cetrimide (CTAB; 99%) were purchased from Winlab (UK), while sodium dodecyl sulfate (SDS; 95%), tween-80 and β-cyclodextrin (β-CD) were obtained from El-Nasr Pharmaceutical Chemicals Company (ADWIC) (Abu Zaabal, Egypt). Moreover, acetic acid 96%, sodium acetate and boric acid were also purchased from ADWIC.

— Ethanol (Fisher, UK)

— Acetonitrile and methanol (Tedia, USA).

## 2.3. Standard solutions

The stock solution of 100.00 µg ml$^{-1}$ for ETO, MOX and NAL was prepared separately by dissolving a definite amount of the studied drugs in methanol for ETO and ethanol for both MOX and NAL. The standard working solutions with different concentrations were prepared by appropriate dilution of the stock solution of ETO, MOX and NAL with ethanol (ETO, 10.00 µg ml$^{-1}$; MOX, 4.00 µg ml$^{-1}$; and NAL, 50.00 µg ml$^{-1}$). Subsequently, these solutions were preserved in the fridge at 2°C until used, and it was found that they were stable for 10 days.

## 2.4. Procedures

### 2.4.1. Construction of calibration graphs

Estimation of ETO, MOX and NAL was done by transferring the previously prepared standard solutions of ETO, MOX and NAL into a group of 10 ml volumetric flasks. Accordingly, the final concentration of MOX, ETO and NAL was 0.04–0.40, 0.10–1.00 and 0.50–5.00 µg ml$^{-1}$, respectively, after completing the mark with ethanol then mixing well. The conventional synchronous fluorimetric was applied for MOX estimation, while the first derivative of synchronous fluorimetric ($^{1}$D) was used for ETO and NAL quantification. Conventional synchronous fluorimetric of MOX was monitored at 371 nm, while $^{1}$D synchronous fluorimetric of ETO and NAL was measured at 257 and 273 nm, respectively, in ethanol. The synchronous fluorescence was scanned at $\Delta\lambda$ of 60 nm for both methods. Hence, the design of calibration curves was resolved by graphing the relative synchronous fluorescence intensity (RSFI) for MOX or the peak amplitude of the ($^{1}$D) technique for ETO and NAL versus the final drug concentration in µg ml$^{-1}$.

### 2.4.2. Analysis of MOX, ETO and NAL in their pharmaceutical formulation

Ten Moxavidex® 400 mg tablets for MOX were accurately weighed and then totally crushed. Subsequently, transferring an accurately weighed amount corresponding to 10 mg for MOX into a volumetric flask (100 ml) and adding ethanol up to 70 ml was carried out. Then, sonication for 20 min and adding ethanol to the mark, respectively, were applied. Consequently, the working standard solution for MOX was achieved via filtration of the previous sonicated solution into a volumetric flask (100 ml) and completing to the mark with ethanol. While for ETO, an aliquot equivalent to 10 mg ETO of Etoposide Mylan® solution was transferred into 100 ml volumetric flasks and completed to the mark with methanol. For NAL, another aliquot equivalent to 10 mg NAL of Nalufine® ampoule was transferred into 100 ml volumetric flasks and completed to the mark with ethanol. The working standard solutions for both ETO and NAL were then prepared by transferring the required volume into a 10 ml volumetric flask and completed to mark with ethanol. Eventually, the calibration graph accomplishment for MOX, ETO and NAL was applied as discussed previously, and then derivatization of a correlative regression equation to estimate their concentration in different pharmaceutical formulations.

### 2.4.3. Procedure for simultaneous analysis of MOX, ETO and NAL in spiked human urine

Simultaneous assessment of MOX, ETO and NAL in the spiked human urine was achieved by applying the facile protein precipitation technique. Aliquots of standard solutions (0.08–0.4, 0.2–1 and 1–5 µg ml$^{-1}$) of MOX, ETO and NAL, respectively, were transferred into a set of 10 ml screw-capped centrifuge tubes containing 500 µl of human urine and 3 ml of acetonitrile. A vortex of these centrifuge tubes for 5 min and centrifugation at 3500 r.p.m. for 30 min were done. Then 1 ml of the supernatant was transferred and completed to 10 ml with ethanol. The calibration graph's performance and regression equation derivatization were presented as estimated before.

# 3. Results and discussion

The excitation and emission fluorescence spectra of ETO, MOX and NAL were recorded at 290/320 nm, 292/460 nm and 280/330 nm, respectively (figure 2). Time-saving and improving spectrofluorimetry's selectivity are the main reasons for applying SF ($\Delta\lambda$ of 60 nm) for the determination of ETO, MOX and NAL (figure 3). As illustrated in figure 3, MOX (green colour spectrum) was superimposed with

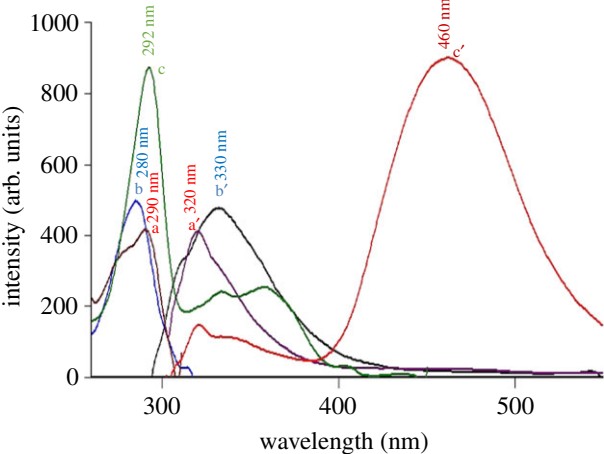

**Figure 2.** Excitation and emission fluorescence spectra of: (a, a′) ETO (1.0 μg ml⁻¹), (b, b′) NAL (4.0 μg ml⁻¹) and (c, c′) MOX (0.08 μg ml⁻¹).

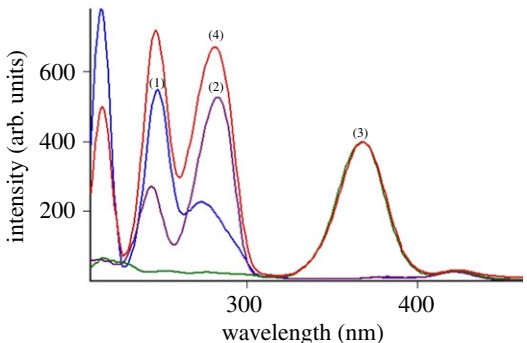

**Figure 3.** Synchronous fluorescence spectra of (1) ETO (1 μg ml⁻¹), (2) NAL (5 μg ml⁻¹), (3) MOX (0.2 μg ml⁻¹), (4) synthetic mixture of ETO (1 μg ml⁻¹), NAL (5 μg ml⁻¹) and MOX (0.2 μg ml⁻¹).

the synchronous fluorescence of the mixture (figure 3), which confirms the direct determination of MOX by direct SF technique at 371 nm without any interferences from the others. ETO exhibits three distinct peaks at 215, 248 and 273 nm, while NAL showed two peaks at 245 and 283 nm. Overlapping between ETO and NAL spectra is obvious, as shown in figure 3, which leads to the addition of the derivative process to improve the selectivity and sensitivity of synchronous scanning in the spectrofluorimetry technique (electronic supplementary material, figures S1–S3).

## 3.1. Experimental conditions optimization

Experimental conditions impacting RSFI of ETO, MOX and NAL were estimated to explore the optimum selectivity and the utmost sensitivity.

### 3.1.1. Effect of diluting solvent

The optimum convenient diluting solvent selection was achieved by involving various solvents, including, water, acetonitrile, ethanol and methanol. Subsequently, the influence of these diluting solvents was varied regarding the drug. For MOX, the maximum RSFI was achieved upon using ethanol, then methanol, and decreased by acetonitrile then water. By contrast, for ETO and NAL, the optimum RSFI was presented by acetonitrile then ethanol, and decreased by methanol then water. Acetonitrile was not selected as it decreased RSFI of MOX, but ethanol showed adequate RSFI of ETO, MOX and NAL (figure 4). Moreover, ethanol resulted in achieving reproducible calibrated results and adding the significance of being an environmentally friendly solvent.

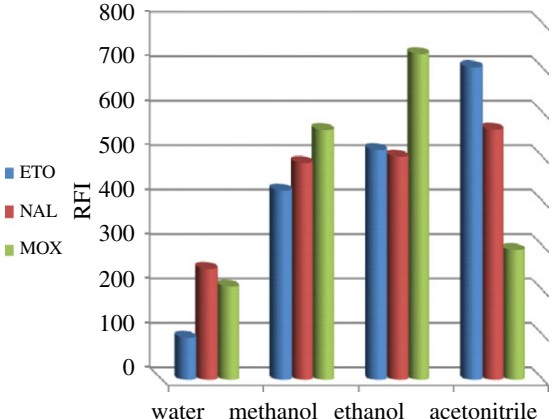

**Figure 4.** Effect of different solvents on the synchronous fluorescence intensity of ETO (1 µg ml$^{-1}$), NAL (5 µg ml$^{-1}$) and MOX (0.4 µg ml$^{-1}$).

### 3.1.2. pH effect

Studying the pH effect on RSFI of ETO, MOX and NAL was carried out by considering a wide range of pH involving 0.2 M acetate buffer (pH 3.0–5.0) and 0.2 M borate buffer (pH 6–10). Different buffers' performance was studied; for MOX, the RSFI was influenced by increasing the pH up to pH 6–7, then decreased by increasing the pH and precipitation was observed at higher pHs (greater than 8). This increase in RSFI at pH up to 6–7 was still lower than the value compared to the absence of buffer. While for ETO and NAL, RSFI was not affected by different buffers (electronic supplementary material, figure S4). So, the proposed method was performed without using a buffer, and this added another value to this work.

### 3.1.3. Surface active agent effect

Consideration of the effect of different surface active agents on RSFI was studied by including 1.0% aqueous solution of SDS, CTAB, Tween-80 and β-CD macromolecules. From the outcomes, the RSFI of ETO, MOX and NAL was higher in the absence of any surfactant (electronic supplementary material, figure S5). Therefore, these surfactants were not involved in these developed approaches.

### 3.1.4. Selection of optimum Δλ

Δλ is a fundamental criterion as it enhances the sensitivity, symmetry of the spectra and resolution, so selecting optimum Δλ was involved in the proposed approaches for synchronous SF quantification of a ternary mixture of ETO, MOX and NAL. Thus, a wide range of Δλ (20–200 nm) was scrutinized. Δλ of 60 nm presented appropriate spectra graphs and convenient ETO and NAL sensitivity. Eventually, Δλ of higher value resulted in irregularities in the spectra graphs, so Δλ of 60 nm was the optimum (electronic supplementary material, figure S6).

## 3.2. Validation of the developed approaches

The proposed synchronous SF quantification of a ternary mixture of ETO, MOX and NAL were validated and affirmed by applying ICH rules [45].

### 3.2.1. Linearity

Designation of calibration graphs was achieved by graphing RSFI for MOX or the peak amplitude of the ($^1$D) technique for ETO and NAL over the concentration ranges 0.04–0.40, 0.10–1.00 and 0.50–5.00 µg ml$^{-1}$, respectively (table 1). Accordingly, correlation coefficients with high values and negligible intercepts values have affirmed the linearity [46]. Moreover, the results of the regression equations are summarized in table 1.

**Table 1.** Analytical performance data for the proposed approach.

| parameter | synchronous fluorimetry MOX at 371 nm | first derivative synchronous fluorimetry | |
| --- | --- | --- | --- |
| | | ETO at 257 nm | NAL at 273 nm |
| linearity range ($\mu$g ml$^{-1}$) | 0.04–0.4 | 0.1–1 | 0.5–5 |
| intercept ($a$) | 26.471 | 1.050 | −0.956 |
| slope ($b$) | 1782 | 30.129 | 5.229 |
| correlation coefficient ($r$) | 0.9999 | 0.9998 | 0.9998 |
| s.d. of residuals ($S_{y/x}$) | 3.19 | 0.22 | 0.21 |
| s.d. of intercept ($S_a$) | 2.48 | 0.17 | 0.16 |
| s.d. of slope ($S_b$) | 10.22 | 0.29 | 0.05 |
| percentage relative standard deviation, % RSD | 0.80 | 1.56 | 1.61 |
| percentage relative error, % Error | 0.33 | 0.64 | 0.66 |
| detection limit, DL (ng ml$^{-1}$) | 4.60 | 19 | 102.90 |
| quantitation limit, QL (ng ml$^{-1}$) | 13.90 | 57.50 | 311.80 |

### 3.2.2. Quantitation limit and detection limit

Considering ICH rules [45], detection limit (DL) and quantitation limit (QL) were mathematically determined using the following equations:

$$\mathrm{DL} = \frac{3.3 S_a}{b} \text{ and } \mathrm{QL} = \frac{10 S_a}{b}.$$

Table 1 abridges the recorded values of DL and QL for ETO, MOX and NAL.

### 3.2.3. Accuracy and precision

Screening the accuracy of the proposed approaches was assessed by carrying a comparison between the developed methods for monitoring the studied drugs with their official and reported ones. The official United States Pharmacopeia (USP) chromatographic method for ETO was performed by dissolving ETO in acetonitrile, separation on phenyl column as a stationary phase and acetate buffer : acetonitrile (74 : 26) as a mobile phase and UV detector at 254 nm [6]. While the SF procedures for MOX and NAL were based on oxidization by Ce(IV) and monitored the resulting Ce(III) at (250/352) nm [22,31], respectively.

A ternary laboratory mixture of ETO, MOX and NAL in different ratios was assessed as well as in their pharmaceutical dosage forms by the proposed approaches and the obtained results are summarized in tables 2 and 3. Ultimately, the calculated Student's $t$-test and variance ratio $F$-test [46] offered no considerable variation in the performance between the current green approaches for monitoring the studied drugs and their official and reported methods regarding accuracy and reproducibility, respectively.

Screening the precision of the developed approaches was affirmed by examining intraday and interday precision. The standard solutions of the studied drugs were determined by using three concentrations in three replicates. For intraday precision it was performed within the same day, but for interday precision was achieved at three sequent days. Table 4 is essentially representing the very low value of % RSD < 2.0%, which confirms the repeatability and reproducibility of the proposed methods.

### 3.2.4. Selectivity

The selectivity of these innovative green techniques was carried out by analysing ETO, MOX and NAL in their pharmaceutical dosage forms. The outcomes are abridged in table 4. The high percentage recovery in addition to lower %RSD (less than 2%) affirming no interference from excipients of pharmaceutical dosage forms for these drugs. Moreover, synchronous SF quantification of these drugs was applied in biological fluids because of the tremendous synchronous selectivity and sensitivity (table 5).

**Table 2.** Application of the proposed approach for the assessment of synthetic mixtures. N.B. Each result is the average of three separate determinations.

| sample | amount taken (µg ml⁻¹) | | | amount found (µg ml⁻¹) | | | % found | | | comparison method/USP [6,22,31] % found | | |
| --- | --- | --- | --- | --- | --- | --- | --- | --- | --- | --- | --- | --- |
| | MOX | ETO | NAL | MOX | ETO | NAL | MOX at 371 nm | ETO at 257 nm | NAL at 273 nm | MOX | ETO | NAL |
| MOX, ETO and NAL synthetic mixture | 0.40 | 1.00 | 5.00 | 0.397 | 0.999 | 5.050 | 99.25 | 99.90 | 101.00 | 98.88 | 100.02 | 98.36 |
| | 0.20 | 1.00 | 5.00 | 0.201 | 1.003 | 4.998 | 100.50 | 100.30 | 99.96 | 101.11 | 99.96 | 100.81 |
| | 0.32 | 0.80 | 4.00 | 0.319 | 0.795 | 4.022 | 99.69 | 99.38 | 100.50 | 99.09 | 98.63 | 99.08 |
| | 0.24 | 0.60 | 3.00 | 0.242 | 0.603 | 2.970 | 100.83 | 100.50 | 99.00 | | | |
| $(\bar{x})$ | | | | | | | 100.07 | 100.02 | 100.12 | 99.69 | 99.53 | 99.42 |
| ±s.d. | | | | | | | ±0.73 | ±0.49 | ±0.86 | ±1.23 | ±0.79 | ±1.26 |
| % RSD | | | | | | | 0.73 | 0.49 | 0.86 | | | |
| % error | | | | | | | 0.36 | 0.24 | 0.43 | | | |
| $t^a$ | | | | | | | 0.51 | 1.01 | 0.88 | | | |
| $F^a$ | | | | | | | 2.88 | 2.53 | 2.16 | | | |

aThe tabulated $t$ and $F$-values at $p = 0.05$ are 2.57 and 9.55, respectively [46].

**Table 3.** Application of the proposed approaches for the assessment of the pharmaceutical dosage forms.

| sample | amount taken ($\mu g\ ml^{-1}$) | amount found ($\mu g\ ml^{-1}$) | % found | comparison method/USP [6,22,31] % found |
|---|---|---|---|---|
| moxavidex 400 mg tablets (436.8 mg MOX/tablet) #180892A | 0.24 | 0.242 | 100.83 | 99.98 |
| | 0.32 | 0.318 | 99.38 | 98.73 |
| | 0.40 | 0.395 | 98.75 | 100.20 |
| $\bar{x} \pm$ s.d. | | | 99.65 ± 1.07 | 99.64 ± 0.79 |
| t | | | 0.02 (2.78)[a] | |
| F | | | 1.81 (19)[a] | |
| nominal content | 398.60 mg MOX | | | |
| etoposide Mylan solution (20 mg ETO/ml) #1077 | 0.60 | 0.602 | 100.33 | 100.13 |
| | 0.80 | 0.796 | 99.50 | 99.23 |
| | 1.00 | 1.004 | 100.40 | 100.21 |
| $\bar{x} \pm$ s.d. | | | 100.08 ± 0.50 | 99.86 ± 0.54 |
| t | | | 0.52 (2.78)[a] | |
| F | | | 1.18 (19)[a] | |
| nominal content | 20.02 mg ETO | | | |
| nalufin 20 mg ampoule (20 mg NAL/ampoule) #195838A | 3.00 | 3.020 | 100.67 | 99.29 |
| | 4.00 | 3.988 | 99.70 | 101.03 |
| | 5.00 | 5.008 | 100.16 | 100.02 |
| $\bar{x} \pm$ s.d. | | | 100.18 ± 0.48 | 100.11 ± 0.87 |
| t | | | 0.11 (2.78)[a] | |
| F | | | 3.24 (19)[a] | |
| nominal content | 20.04 mg NAL | | | |

[a]The tabulated t- and F-values at $p = 0.05$ [46].

## 3.3. Applications

### 3.3.1. Estimation of ETO, MOX and NAL in pharmaceutical dosage forms

SF in derivative form was performed for sensitive quantification of ETO in its solution form in comparing with a USP chromatographic study [6]. Where SFI in conventional and ($^1$D) techniques was applied for selective estimation for MOX and NAL in their tablets and ampoules in comparison with a SF procedure [22,31], respectively. Afterward, the results are explained in table 3. Furthermore, the accuracy and precision of this comparison were confirmed regarding Student's t-test and variance ratio F-test [46]. So, no significant variance between these developed green techniques and the USP method for ETO and the SF procedure for MOX and NAL.

### 3.3.2. Simultaneous analysis of ETO, MOX and NAL in spiked human urine

ETO is excreted in urine as unchanged (30–50% of a dose) within 72 h, and its volume of distribution for adults was (7–17) $l\ m^{-2}$ [47]. MOX bioavailability is 90%, and it is excreted in the urine unchanged [47]. NAL is rapidly absorbed after intramuscular injection and excreted unchanged [47]. Due to the remarkable synchronous selectivity and sensitivity, the RSFI quantification of ETO, MOX and NAL was applied in spiked biological fluids. Subsequently, the obtained results are illustrated in figure 5 and abridged in table 5. A linear relationship was obtained by drawing the RSFI for MOX or the peak amplitude of the ($^1$D) technique for ETO and NAL versus the final drug concentration ($\mu g\ ml^{-1}$) and the least square linear regression equations are as follows:

SCF in conventional form

$$\text{RSFI} = 1804C - 17.65 \quad (r = 0.999) \text{ for MOX}$$

**Table 4.** Precision details for assessing MOX, ETO and NAL drugs.

| amount taken (μg ml$^{-1}$) | % found ± % RSD | % error |
|---|---|---|
| synchronous fluorimetry | | |
| MOX at 371 nm | | |
| intraday; 0.16 | 100.32 ± 0.43 | 0.25 |
| 0.24 | 99.66 ± 0.24 | 0.14 |
| 0.32 | 100.22 ± 0.56 | 0.32 |
| interday; 0.16 | 100.66 ± 1.21 | 0.70 |
| 0.24 | 99.62 ± 0.85 | 0.49 |
| 0.32 | 99.88 ± 1.06 | 0.61 |
| first derivative synchronous fluorimetry | | |
| ETO at 257 nm | | |
| intraday; 0.40 | 98.98 ± 0.42 | 0.24 |
| 0.60 | 99.86 ± 0.34 | 0.20 |
| 0.80 | 99.33 ± 0.83 | 0.48 |
| interday; 0.40 | 98.18 ± 1.00 | 0.58 |
| 0.60 | 98.66 ± 1.07 | 0.62 |
| 0.80 | 99.76 ± 1.33 | 0.77 |
| NAL at 273 nm | | |
| intraday; 2.00 | 99.56 ± 0.92 | 0.53 |
| 3.00 | 100.14 ± 0.36 | 0.21 |
| 4.00 | 99.87 ± 0.63 | 0.36 |
| interday; 2.00 | 99.01 ± 0.79 | 0.46 |
| 3.00 | 98.32 ± 1.11 | 0.64 |
| 4.00 | 98.24 ± 0.86 | 0.49 |

**Table 5.** MOX, ETO and NAL simultaneous assay results in spiked human urine.

| spiked urine sample | amount taken (μg ml$^{-1}$) | | | amount found (μg ml$^{-1}$) | | | % found | | |
|---|---|---|---|---|---|---|---|---|---|
| | MOX | ETO | NAL | MOX | ETO | NAL | MOX at 371 nm | ETO at 257 nm | NAL at 273 nm |
| MOX, ETO and NAL | 0.08 | 0.20 | 1.00 | 0.075 | 0.167 | 0.855 | 95.00 | 83.50 | 85.50 |
| | 0.16 | 0.40 | 2.00 | 0.164 | 0.429 | 2.171 | 102.50 | 107.25 | 108.55 |
| | 0.32 | 0.80 | 4.00 | 0.327 | 0.845 | 4.069 | 102.19 | 105.63 | 101.73 |
| | 0.40 | 1.00 | 5.00 | 0.393 | 0.959 | 4.906 | 98.25 | 95.90 | 98.12 |
| ($\bar{x}$) | | | | | | | 99.49 | 98.07 | 98.48 |
| ±s.d. | | | | | | | ±3.56 | ±10.93 | ±9.67 |
| % RSD | | | | | | | 3.58 | 11.15 | 9.82 |
| % Error | | | | | | | 1.79 | 5.57 | 4.91 |

SCF in the ($^1$D) technique

$$(^1D) = 32.985C - 4.286 \quad (r = 0.993) \text{ for ETO}$$

$$(^1D) = 4.673C + 8.076 \quad (r = 0.997) \text{ for NAL.}$$

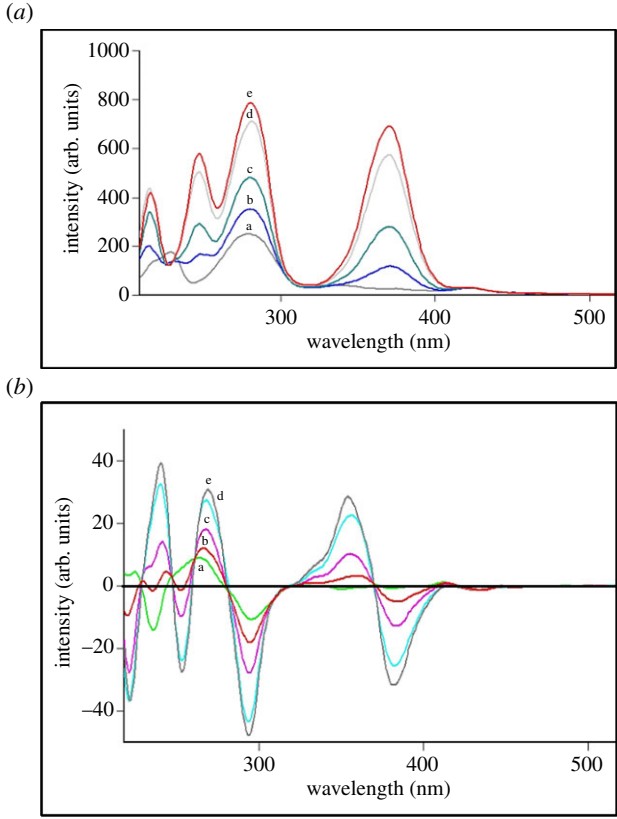

**Figure 5.** Application of: (*a*) synchronous spectrofluorimetry and (*b*) first derivative synchronous spectrofluorimetry for the determination of the studied drugs in spiked human urine: (a) blank urine; (b) ETO (0.2 μg ml$^{-1}$), NAL (1 μg ml$^{-1}$) and MOX (0.08 μg ml$^{-1}$); (c) ETO (0.4 μg ml$^{-1}$), NAL (2 μg ml$^{-1}$) and MOX (0.16 μg ml$^{-1}$); (d) ETO (0.8 μg ml$^{-1}$), NAL (4 μg ml$^{-1}$) and MOX (0.32 μg ml$^{-1}$); (e) ETO (1 μg ml$^{-1}$), NAL (5 μg ml$^{-1}$) and MOX (0.4 μg ml$^{-1}$).

## 3.4. Assessment of greenness of synchronous fluorimetric approaches

Evaluation of the analytical procedure designed as green or not was involved as the main feature today. Various parameters were included for this assessment, such as NEMI [41]. NEMI is a facilely legible pictogram that incorporates four coloured sections, either white or green, based on the identical criterion; pH (2–12), hazardous waste, final waste volumeless than 50 g or l, and no implication of the involved reagent in the list of persistent, bioaccumulative and toxic chemicals [48,49]. Therefore, the results were assessed by mentioned criteria and affirmed the greenness of SF (table 6). Another assessment feature to overcome the drawbacks of NEMI [41] is the semi-quantitative technique, analytical eco-scale. This eco-scale technique is proved by calculating penalty points to any parameter that discrepancy with idealistic green design counted on reagents amount involved and its hazardousness, energy used and waste production, and then subtracting from ideal value 100 [41]. Globally harmonized system (GHS) was established to estimate the reagent's hazards by labelling each reagent by (1–9) graphic pictograms. GHS was categorized and characterized each chemical as danger when its penalty point is two and warning when it is one. Finally, the analytical eco-scale score was 86, exhibiting excellent greenness (table 6) [41]. GAPI has been recently established as an assessment feature for evaluating the greenness of the SF technique. GAPI is the perfect scheme as it diminishes the difficulties found with other techniques as it relies on a pictogram that classifies the greenness of each stage of the SF, including a colour scale of three levels: green, yellow or red. The red domain of 1 and 15 referred to no treatment for off-line sampling and waste. The yellow domain of 5 and 14 exhibiting the easy sample preparation method, and the waste of 10 ml were generated. By applying GAPI for assessment of the studied drugs in spiked human urine, red colour for the micro-extraction is recorded due to the use of acetonitrile, a non-green solvent.

This graphical presentation indicated that the SF was considered a green technique with minimal effect on human health and the environment as comprehensive all GAPI features (table 6).

**Table 6.** Results for evaluation of greenness of the proposed approaches.

*1. NEMI pictogram*

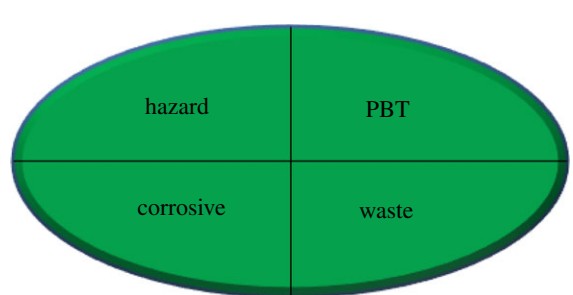

*2. analytical eco-scale score*[a]

| Item | no of pictogram | word sign | penalty points |
|---|---|---|---|
| reagent; volume (ml) ethanol; 10 ml | 2 | danger | 8 |
| spectrofluorimeter | | | 0 |
| occupational hazard | | | 0 |
| waste | | | 6 |
| total penalty points | | | 14 |
| analytical eco-scale score | | | 86 |

*3. Green analytical procedure index (GAPI)*

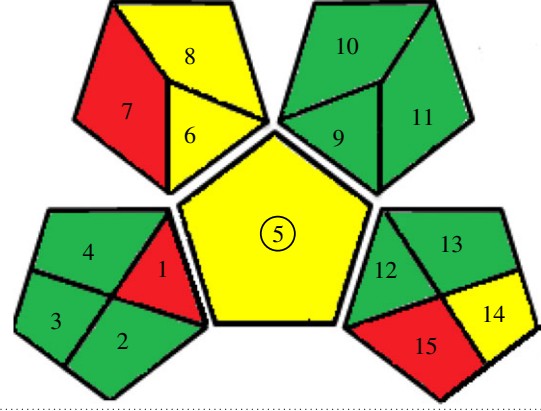

[a]Eco-scale reference range: optimum green 100, excellent greater than 75, acceptable greater than 50 and inadequate less than 50.

Consequently, green methods for synchronous SF quantification of ETO, MOX and NAL in their pharmaceutical dosage forms and biological fluids have been advanced and validated with the least laboratory requirements.

## 4. Conclusion

Unique green approaches for synchronous SF quantification of ETO, MOX and NAL in their synthetic laboratory mixtures, pharmaceutical dosage forms and biological fluid were designed. The scope of this study is to develop an advantageous sustainable method used for therapeutic drug monitoring of the studied drugs. The proposed green SF approaches were distinguished by their feasible method: low time consumed, high quality due to its sensitivity and economic impact. In addition, the green SF approaches could be easily employed in routine work performed in quality control laboratories.

Data accessibility. Data are available from the Dryad Digital Repository: https://dx.doi.org/10.5061/dryad. gmsbcc2n6 [50].

The data are provided in electronic supplementary material [51].

Authors' contributions. M.M.T. carried out the laboratory work, participated in data analysis and participated in the design of the study; F.B. coordinated the study, participated in data analysis and helped draft the manuscript; M.M.S. drafted the manuscript, carried out the statistical analyses, conceived of the study and designed the study. All authors gave final approval for publication.

Competing interests. We declare we have no competing interests.

Funding. We received no funding for this study.

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
