## [Peer Review File · Royal Society Open Science]

Review History

RSOS-210683.R0 (Original submission)

Review form: Reviewer 1

Is the manuscript scientifically sound in its present form?

Yes

Are the interpretations and conclusions justified by the results?

Yes

Is the language acceptable?

Yes

Do you have any ethical concerns with this paper?

No

Have you any concerns about statistical analyses in this paper?

No

Recommendation?

Accept with minor revision (please list in comments)

Comments to the Author(s)

The manuscript# RSOS-210683 entitled " An eco-friendly fluorimetric approaches for the simultaneous estimation of antineoplastic, fluoroquinolone, and opioid analgesic drugs in human urine" is of interest and suitable for the scope of Royal Society Open Science Journal.

The authors present a spectrofluorimetric assay for the determination of three drugs, etoposide, moxifloxacin and nalbuphine, in standard solutions, in dosage forms as well as in spiked human urine. The assay relies on conventional and first derivative synchronous fluorimetric of the ethanolic solution of the ternary mixture. The manuscript is clear and well written and may be published after a minor revision and considering the points listed on the author's side.

1. The article needs careful revision and editing.
2. The title should be changed to be more representative.
3. The sensitivity of the Agilent device was sited at 800 v..... The mode of operation should be written instead.
4. Under the procedure section, the Statement "completing the volume" must be changed to be " completing the mark."
5. In page# 4....."Subsequently, these solutions were preserved in the fridge until utilized" ... at which degree?
6. In procedure section: it is better to use the word mix instead of blend. Blending usually when the biological tissue is incorporated in the method.
7. In Result and discussion section: why the authors used to number the figures in the zero non decimal way? please use the ordinary numbering formate.
8. In experimental optimization section:
 - The abbreviation of Surface Active Agent (SAG) should be changed to SAA.
 - Also "Therefore, these surfactants were not involved in this innovatory green process.....so what is the sence of trying this effect?
9. The dosage forms nominal content should be calculated, and the statistical obtained data should be added in Table #4.
10. Figure #1: the image resolution should be improved....
11. Figure #2 Please, point out the excitation and emission wavelengths used.
12. Figures#4,5,6,8,9 should be transferred to the supplementary material data file.
13. The intersection and symbols in figures 6 and 10 was displaced so need to be corrected.

Review form: Reviewer 2 (Ahmed M. Abdel-Megied)

Is the manuscript scientifically sound in its present form?

Yes

Are the interpretations and conclusions justified by the results?

Yes

Is the language acceptable?

No

Do you have any ethical concerns with this paper?

No

Have you any concerns about statistical analyses in this paper?

No

Recommendation?

Major revision is needed (please make suggestions in comments)

Comments to the Author(s)

The manuscript presents two synchronous spectrofluorimetric methods for the determination of etoposide, moxifloxacin, and nalbuphine in dosage forms and spiked human urine. This manuscript has several drawbacks and I have the following comments and concerns.

Recommendation: Major revision and English editing is a must.

Comments:

General comments:

1. This paper was not well written or revised, very poor English, many grammatical and spelling mistakes, and unclear descriptions. This paper can't be accepted in this form and English editing by a professional or a native English speaker is a must.
2. The number of figures should be reduced and the resolution of the figures should be improved (at least 300 dpi).
3. Please revise the significant figures in tables and all over the manuscript.
4. Some references don't follow the journal style and the journal names need to be abbreviated. For example, Ref. no. 11, 12, 13, 16, 18, 23, 24, 26, 30, 32, 33, 35.
5. Please revise the adjustment of the text within the manuscript to be uniform including the references.
6. The statement "innovatory green processes" was unnecessarily repeated many times throughout the manuscript. Can you explain why?

Specific comments:

1. Title: The title of any paper is an important indication of its quality, so it should be correct and well revised.

- Please, remove "An" from the title as it is incorrect where "approaches" is a plural, not a singular word.

- Please, replace drug classes in the title with the name of actually studied drugs.

- In the title, the authors mentioned that the method was applied to human urine. However, only spiked urine appeared in the manuscript and no real urine was analyzed. Analysis of spiked urine is not a guarantee for the successful application in real samples. So the authors should try to analyze real urine, otherwise, the urine should be removed from the title.

2. Abstract: It contains many mistakes and should be rewritten and revised well.

For example:

- Line no. 12, Change "leukemia cancer" into "leukemia" in the abstract and through the manuscript.

- Line no. 19 "Consequently, innovatory green processes for synchronous spectrofluorimetric quantification of a ternary mixture of antineoplastic, fluoroquinolone, and opioid analgesic drugs", Where is the sentence?. Please, write a complete sentence. There is no sentence without a verb.
- Line no. 25, Change "Conventional synchronous fluorimetric of MOX" into "Conventional synchronous fluorimetric determination of MOX".
- Line no. 27, change "first derivative synchronous fluorimetric of ETO and NAL" into " first derivative synchronous fluorimetric determination of ETO and NAL".
- Please, change the word "processes" into "methods" in the abstract and through the manuscript.
- Line no. 33, "The outcomes of these designed environmentally friendly processes, a linear correlation was described in the ranges of 0.04-0.40, 0.1-1.0, and 0.5-5.0 µg/ mL for MOX, ETO, and NAL, respectively. This sentence is confusing and should be modified.
- Line no. 36, "Accordingly, several pharmaceutical dosages were achieved within the range simply and precisely". What do you mean by "achieved"? Rewrite this sentence.
- Line no. 40, please replace "biological fluid" with "spiked human urine samples" through the manuscript and add some details in the abstract about the main results for the determination of the studied drugs in spiked human urine.
- Line no. 43, change " roles" into "rules". Its spelling is incorrect. This is a repeated mistake, correct it through the manuscript.
- Line no. 45, the abstract should not contain abbreviations such as NEMI and GAPI.
- Line no. 48, Keywords, change "Synchronous fluorimetric" into "Synchronous fluorimetry".

3. Introduction:

- Advantage of the proposed methods over the published ones should be introduced in the introduction.

The Introduction contains many writing mistakes:

- Line no. 6, "and is categorized as alkaloid 9-[4, 6-O-(R)-ethylidene- β -D-glucopyranoside]; (5S, 5aR, 8aS,9R)-9-(4,6-O-Ethylidene- β -D-glucopyranosyloxy)-5, 8, 8a, 9-tetra hydro -5-(4-hydroxy-3,5-dimethoxy phenyl)-isobenzofuro[5,6 f] [1,3] benzodioxol-6(5aH)-one [1-3]", This sentence is unclear, rewrite it.
- Line no. 13, change "MOX is named" into ""MOX is named as".
- Line no. 21, "USP pharmacopeia", correct it into "USP" or "United States pharmacopeia".
- Lines no. 23-38, rewrite this paragraph and make it simpler.
- Lines 39-42, "Hence, innovatory green processes for synchronous spectrofluorimetric (SF)

quantification of a ternary mixture of ETO, MOX, and NAL in their pharmaceutical dosage form and/or biological fluids". Where is the sentence?. Please, write a complete sentence. There is no sentence without a verb.

- Line no. 45, "Since the selectivity of spectrofluorimetry is improved by applying synchronous scanning [39]", where is the rest of the sentence?.

-Line no. 47, change " the significance of" into "The significance of".

-Line no. 49, what do you mean by "is a manifestation of selectivity and sensitivity". Please rewrite it.

- Page 3, line no. 1, add "an" before "additional".

- Pages 3, line no. 2, "NEMI, GAPI", define the two abbreviations at their first appearance in the manuscript.

-Page 3, line no. 11, "The scope", I think this word is inappropriate, "The aim", "The objective", or "The goal" may be more appropriate.

- Page 3, line no. 19, change "roles" into "rules". The spelling is incorrect.

4. Experimental:

- Section 2.1:

- Please, add all details about the manufacturer of instruments i.e. the city and country....etc.

- The used cell should be written.

- Line no. 25, change "Fluorescence" into "synchronous fluorescence spectra".

- Line no. 28-30, "Thermostatically controlling and shaking for all experimental features was carried out by utilizing the England water bath of Cambridge Ltd". You didn't study the effect of temperature, what do you mean by this sentence?.

-Line no. 33, "and separation using centrifuge model 2-16P (Germany)", correct this sentence.

- Section 2.2:

-Lines no. 39-44, use the abbreviations for the drugs as they previously mentioned in the "Introduction" section.

- Line no. 53, change "the output of Future Pharmaceutical Industries" into "the product of Future Pharmaceutical Industries".

-Page 4, line no. 8, change "The surfactant" into "The surfactants".

- Section 2.3:

- Please revise the significant figures in this section and all over the manuscript

- Line no. 20, replace the word "performed" as it is inappropriate.

-Line no. 23, remove the word " Hence".

- Line no. 26, change "where ETO was 10µg/mL, MOX was 4 µg/mL and NAL was 50 µg/mL". rewrite this sentence.

- Section 2.4.1:

- Page 5, lines no. 1- 21: under the title "Analysis of MOX, ETO, and NAL in their pharmaceutical formulation:" and page 8, lines 6 - 23, the authors described the analysis of separate dosage forms but not simultaneous determination. This is not matched with the title of the manuscript.

- Line no. 37, replace the sentence "was to the extent of" as it is inappropriate

-Lines no. 41-43, "Two eco-friendly synchronous fluorimetric processes (conventional and first derivative) were applied to estimate these drugs", there is no need for this sentence.

- Line no. 46, "where 1D", the word "where" here is not appropriate, replace it with a suitable word.

- Section 2.4.2:

- Line no. 3, "Moxavidex® 400 mg tablets for MOX were accurately weighed and then utterly crushed". The authors didn't clarify the number of tablets included in this study.

- Line no. 6, "Therefore, sonication", therefore here is not appropriate please replace it.

The authors used "Hence", "Therefore", "consequently" many times incorrectly at the beginning of the sentences, please, revise this through the manuscript.

- Line no. 8, "sonication for 20 min", please, explain why 20 min sonication was necessary. I think it is too much time for sonication.

- Line no. 10, "was implemented", replace this verb as it is not appropriate.

- Lines no. 11-22, rewrite this paragraph as it contains many mistakes and unclear descriptions.

- Lines no. 11-18, "While for ETO and NAL, an amount equivalent to 10 mg ETO of Etoposide Mylan® solution and 10 mg NAL of Nalufine® ampoule were individually moved into 100 mL volumetric flasks and completing to the mark with methanol and ethanol, respectively. Additionally, the achievement of a working standard solution for both ETO and NAL was applied by adding ethanol to the final mark". There is a contradiction in this paragraph. Please, clarify how you prepared the working solutions for ETO and NAL.

- Section 2.4.3:

- The authors described the analysis of each drug separately in spiked human urine but not simultaneous determination. This is not matched with the title of the manuscript. The authors must study the simultaneous determination of different mixtures with different ratios of the studied drugs in spiked human urine, not each drug alone. This is the main objective of the study as claimed by the authors.

- Lines no. 25-27, "Synchronous assessment of MOX, ETO, and NAL in the biological fluid was attained by involving human urine to design a standard calibration curve", this sentence is confusing and should be modified.

- Line no. 32, "and 3 mL of acetonitrile into a set of covered tubes", was this volume optimized? I think it is too much with respect to the greenness of the method. Why did you use acetonitrile not ethanol especially it is the diluting solvent in your study?.

- What do you mean by "covered tubes"? You should also mention their volumes.

5. Results and discussion:

- Page 5, Lines no. 40-51, the authors referred only to the figures and didn't explain in detail the synchronous spectrofluorimetric determination of the studied drugs.

- Where is the figure for the synchronous spectrofluorimetric determination of MOX?.

- Figures 2 and 3, MOX ($0.08 \mu\text{g/mL}$) in Figure 2 gave fluorescence intensity of about 900, while in Fig. 3 when its concentration increased to $0.2 \mu\text{g/mL}$, the synchronous fluorescence intensity was about 400. It is well known that the synchronous spectrofluorimetry enhances the sensitivity not decreases it. Please, explain this contradiction.

- Figure 3, There is a big difference between the SF intensity when the drugs were determined simultaneously in a synthetic mixture and when each drug was determined separately especially for ETO and NAL. The two spectra (i.e in the synthetic mixture and separately) should give the same SF intensity.

- Remove "A" from the beginning of Figure 2 caption.

- Lines no. 42-44, "As illustrated in Fig. 02, MOX could be determined by the conventional spectrofluorimetric technique". You determined MOX using the synchronous spectrofluorimetric technique, not conventional spectrofluorimetry.

- Figures 4-6, please add the blank spectra to the figures.

- Page 6, line no. 3, "Experimental conditions impacting RFI", you didn't measure the RFI, rather you measured synchronous fluorescence intensity, please, correct this in section 3.1 and through the manuscript.

- Figure 7 caption, "Effect of different solvent", correct "solvent" into "solvents".

- Page 6, line no. 23, "Studying the pH effectiveness", replace "effectiveness" as it is inappropriate.

- Page 6, line no. 38, "1.0 % aqueous solution of SDS, CTAB, Tween-80, and β -CD macromolecules". Explain why the studied surfactants were used in this concentration (1.0 %)?.

- Figure 9 caption, "Effect of different surfactant", correct "surfactant" into "surfactants".

- Figures 7-9 can be collected in one figure.

- Section 3.1.4., please provide the figure for studying the effect of $\Delta \lambda$.

- Page 7, line no. 8, "applying ICH roles", change "roles" into "rules". The spelling is incorrect.

- Table 1, "Quantitation Limit of, QL (ng/mL)", remove "of".
- Section "3.2.3. Accuracy and precision", this section contains many mistakes and unclear descriptions. It should be revised and rewritten.
- The results of the proposed methods should be statically compared with the reported or reference methods to confirm the accuracy and precision in both pure forms and dosage forms. The authors compared them only in the dosage forms.
- Page 7, lines no. 28-30, "Accuracy of innovatory green synchronous spectrofluorimetric quantification of a ternary laboratory mixture of ETO, MOX, and NAL was applied". This sentence is confusing and should be modified.
- Page 7, line no. 30, "This performance was carried by", correct this sentence.
- Page 7, Lines no. 35-39, "Table 2 showed the accuracy of innovatory green synchronous spectrofluorimetric in conventional and derivative form for estimation of these coadministered drugs, and Fig. 09 confirmed this", How Figure 09 confirmed this?. Figure 09 is for the effect of different surfactants.
- Page 7, line no. 39, "Assessment precision of", correct this sentence.
- Page 7, line no. 42, "the consequence was abridged (Table 3), which approving excellent precision". This sentence is confusing and should be modified.
- Table 3, Change "% Found" in the heading of the table into "% Found \pm % RSD" and there is no need for the column of % RSD.
- Table 4, add the tabulated t and F values for Moxavidex tablets and Etoposide Mylan solution.
- Page 8, line no. 13, "and UV-detector was used for monitoring was at 254 nm". Revise this sentence.
- Page 8, section "3.3.1. Estimation of ETO, MOX, and NAL in pharmaceutical dosage forms". This section contains many mistakes and unclear descriptions; it should be revised well and modified.
- Page 8, line no. 30, "NAL is rapidly absorbed after intramuscular and is excreted unchanged". Revise this sentence.
- Page 8, line no. 31, "the SCF", this abbreviation was not defined before in the manuscript.
- Page 8, line no. 33, "Subsequently, the resulted were presented in (Fig. 10)", revise and correct this sentence.
- Page 8, lines no. 33-37, "A linear relationship by graphing (RFI) for MOX or the peak amplitude of the (1D) technique for ETO and NAL versus the final drug concentration ($\mu\text{g}/\text{mL}$) was approved". Rewrite this sentence.
- Page 8, lines no. 49-51, "Lastly, the monitoring of ETO, MOX, and NAL in human urine was involved in Table 5 and derivatized by utilizing the previous regression equations". This sentence is confusing and should be modified.

- Figure 10:

- I think the caption is not related to the provided figure. The authors wrote "(a) Blank plasma", the study didn't use any plasma.

- "(A) synchronous spectrofluorimetry (B) first derivative synchronous spectrofluorimetry for the determination of the studied drugs in spiked human urine:", Where is "A" and "B" in the provided figure?.

- The figure contains letters "a,b,c,d,e" and you defined a, b, and c only.

- The authors mentioned in the manuscript that they analyzed the studied drugs separately in the spiked human urine and provided the results in Table 5 but in the provided figure they claimed that they determined them as mixtures. There is contradiction between the figure and what explained in the manuscript and Table 5.

- Page 9, line no. 7, "based on concurred with identical criterion", revise and correct.

- Page 9, line no. 11, "Therefore, the results were significantly performed by mentioned criteria", It is better to change "significantly performed" with "assessed" or "evaluated".

- Page 9, line no. 21-24, "as GHS was categorized and characterized every chemical if the chemical penalty points are equal to two and one as a danger" or "warning", respectively." This sentence is confusing and should be corrected.

- Page 9, line no. 27, "Newly, Green Analytical Procedure Index (GAPI) has been established", revise and correct. "Newly" here is not appropriate.

-Page 9, line no. 30, "GAPI is recorded as the best tool", revise and correct.

- Page 9, line no. 37-41, "Furthermore, GAPI has been performed to the SF method involved in estimating the ETO, MOX, and NAL in human urine; it was recorded that yellow color for micro-extraction is applied red color (acetonitrile, a non-green solvent)". This sentence is confusing and should be modified.

- Table 6, in Analytical Eco-Scale, the calculation of penalty points for waste is incorrect. It should be 6, not 3. In this method, the waste was 10 mL = 3 penalty points and no treatment of the waste = 3 penalty points, so the total penalty points for waste should be 6.

- Table 6, you should add the reference ranges of Analytical Eco-Scale to the table footnotes.

6. Conclusion:

- Page 10, lines no. 3-13, this paragraph is confusing and contains many mistakes; it should be revised and modified.

Decision letter (RSOS-210683.R0)

Dear Dr Salim:

Title: An eco-friendly fluorimetric approaches for the simultaneous estimation of antineoplastic, fluoroquinolone, and opioid analgesic drugs in human urine
Manuscript ID: RSOS-210683

The editor assigned to your manuscript has now received comments from reviewers. We would like you to revise your paper in accordance with the referee and Subject Editor suggestions which can be found below (not including confidential reports to the Editor). Please note this decision does not guarantee eventual acceptance.

Please submit your revised paper before 14-Jul-2021. Please note that the revision deadline will expire at 00.00am on this date. If we do not hear from you within this time then it will be assumed that the paper has been withdrawn. In exceptional circumstances, extensions may be possible if agreed with the Editorial Office in advance. We do not allow multiple rounds of revision so we urge you to make every effort to fully address all of the comments at this stage. If deemed necessary by the Editors, your manuscript will be sent back to one or more of the original reviewers for assessment. If the original reviewers are not available we may invite new reviewers.

RSC Associate Editor:
Comments to the Author:
(There are no comments.)

RSC Subject Editor:
Comments to the Author:
(There are no comments.)

Reviewers' Comments to Author:
Reviewer: 1

Comments to the Author(s)

The manuscript# RSOS-210683 entitled " An eco-friendly fluorimetric approaches for the simultaneous estimation of antineoplastic, fluoroquinolone, and opioid analgesic drugs in human urine" is of interest and suitable for the scope of Royal Society Open Science Journal.

The authors present a spectrofluorimetric assay for the determination of three drugs, etoposide, moxifloxacin and nalbuphine, in standard solutions, in dosage forms as well as in spiked human urine. The assay relies on conventional and first derivative synchronous fluorimetric of the ethanolic solution of the ternary mixture. The manuscript is clear and well written and may be published after a minor revision and considering the points listed on the author's side.

1. The article needs careful revision and editing.
2. The title should be changed to be more representative.
3. The sensitivity of the Agilent device was sited at 800 v..... The mode of operation should be written instead.
4. Under the procedure section, the Statement "completing the volume" must be changed to be "completing the mark."
5. In page# 4....."Subsequently, these solutions were preserved in the fridge until utilized" ... at which degree?
6. In procedure section: it is better to use the word mix instead of blend. Blending usually when the biological tissue is incorporated in the method.
7. In Result and discussion section: why the authors used to number the figures in the zero non decimal way? please use the ordinary numbering formate.
8. In experimental optimization section:
 - The abbreviation of Surface Active Agent (SAG) should be changed to SAA.
 - Also "Therefore, these surfactants were not involved in this innovatory green process.....so what is the sence of trying this effect?
9. The dosage forms nominal content should be calculated, and the statistical obtained data should be added in Table #4.
10. Figure #1: the image resolution should be improved....
11. Figure #2 Please, point out the excitation and emission wavelengths used.
12. Figures#4,5,6,8,9 should be transferred to the supplementary material data file.
13. The intersection and symbols in figures 6 and 10 was displaced so need to be corrected.

Reviewer: 2

Comments to the Author(s)

The manuscript presents two synchronous spectrofluorimetric methods for the determination of etoposide, moxifloxacin, and nalbuphine in dosage forms and spiked human urine. This manuscript has several drawbacks and I have the following comments and concerns.

Recommendation: Major revision and English editing is a must.

Comments:

General comments:

1. This paper was not well written or revised, very poor English, many grammatical and spelling mistakes, and unclear descriptions. This paper can't be accepted in this form and English editing by a professional or a native English speaker is a must.
2. The number of figures should be reduced and the resolution of the figures should be improved (at least 300 dpi).
3. Please revise the significant figures in tables and all over the manuscript.
4. Some references don't follow the journal style and the journal names need to be abbreviated. For example, Ref. no. 11, 12, 13, 16, 18, 23, 24, 26, 30, 32, 33, 35.
5. Please revise the adjustment of the text within the manuscript to be uniform including the references.
6. The statement "innovatory green processes" was unnecessarily repeated many times throughout the manuscript. Can you explain why?

Specific comments:

1. Title: The title of any paper is an important indication of its quality, so it should be correct and well revised.

- Please, remove "An" from the title as it is incorrect where "approaches" is a plural, not a singular word.

- Please, replace drug classes in the title with the name of actually studied drugs.

- In the title, the authors mentioned that the method was applied to human urine. However, only spiked urine appeared in the manuscript and no real urine was analyzed. Analysis of spiked urine is not a guarantee for the successful application in real samples. So the authors should try to analyze real urine, otherwise, the urine should be removed from the title.

2. Abstract: It contains many mistakes and should be rewritten and revised well.

For example:

- Line no. 12, Change "leukemia cancer" into "leukemia" in the abstract and through the manuscript.

- Line no. 19 "Consequently, innovatory green processes for synchronous spectrofluorimetric quantification of a ternary mixture of antineoplastic, fluoroquinolone, and opioid analgesic drugs", Where is the sentence?. Please, write a complete sentence. There is no sentence without a verb.

- Line no. 25, Change "Conventional synchronous fluorimetric of MOX" into "Conventional synchronous fluorimetric determination of MOX".

- Line no. 27, change "first derivative synchronous fluorimetric of ETO and NAL" into " first derivative synchronous fluorimetric determination of ETO and NAL".
- Please, change the word "processes" into "methods" in the abstract and through the manuscript.
- Line no. 33, "The outcomes of these designed environmentally friendly processes, a linear correlation was described in the ranges of 0.04-0.40, 0.1-1.0, and 0.5-5.0 µg/mL for MOX, ETO, and NAL, respectively. This sentence is confusing and should be modified.
- Line no. 36, "Accordingly, several pharmaceutical dosages were achieved within the range simply and precisely". What do you mean by "achieved"? Rewrite this sentence.
- Line no. 40, please replace "biological fluid" with "spiked human urine samples" through the manuscript and add some details in the abstract about the main results for the determination of the studied drugs in spiked human urine.
- Line no. 43, change " roles" into "rules". Its spelling is incorrect. This is a repeated mistake, correct it through the manuscript.
- Line no. 45, the abstract should not contain abbreviations such as NEMI and GAPI.
- Line no. 48, Keywords, change "Synchronous fluorimetric" into "Synchronous fluorimetry".

3. Introduction:

- Advantage of the proposed methods over the published ones should be introduced in the introduction.

The Introduction contains many writing mistakes:

- Line no. 6, "and is categorized as alkaloid 9-[4, 6-O-(R)-ethylidene- β -D-glucopyranoside]; (5S, 5aR, 8aS,9R)-9-(4,6-O-Ethylidene- β -D-glucopyranosyloxy)-5, 8, 8a, 9-tetra hydro -5-(4-hydroxy-3,5-dimethoxy phenyl)-isobenzofuro[5,6 f] [1,3] benzodioxol-6(5aH)-one [1-3]", This sentence is unclear, rewrite it.
- Line no. 13, change "MOX is named" into ""MOX is named as".
- Line no. 21, "USP pharmacopeia", correct it into "USP" or "United States pharmacopeia".
- Lines no. 23-38, rewrite this paragraph and make it simpler.
- Lines 39-42, "Hence, innovatory green processes for synchronous spectrofluorimetric (SF) quantification of a ternary mixture of ETO, MOX, and NAL in their pharmaceutical dosage form and/or biological fluids". Where is the sentence?. Please, write a complete sentence. There is no sentence without a verb.
- Line no. 45, "Since the selectivity of spectrofluorimetry is improved by applying synchronous scanning [39]", where is the rest of the sentence?.
- Line no. 47, change " the significance of" into "The significance of".

-Line no. 49, what do you mean by "is a manifestation of selectivity and sensitivity". Please rewrite it.

- Page 3, line no. 1, add "an" before "additional".

- Pages 3, line no. 2, "NEMI, GAPI", define the two abbreviations at their first appearance in the manuscript.

-Page 3, line no. 11, "The scope", I think this word is inappropriate, "The aim", "The objective", or "The goal" may be more appropriate.

- Page 3, line no. 19, change "roles" into "rules". The spelling is incorrect.

4. Experimental:

- Section 2.1:

- Please, add all details about the manufacturer of instruments i.e. the city and country....etc.

- The used cell should be written.

- Line no. 25, change "Fluorescence" into "synchronous fluorescence spectra".

- Line no. 28-30, "Thermostatically controlling and shaking for all experimental features was carried out by utilizing the England water bath of Cambridge Ltd". You didn't study the effect of temperature, what do you mean by this sentence?.

-Line no. 33, "and separation using centrifuge model 2-16P (Germany)", correct this sentence.

- Section 2.2:

-Lines no. 39-44, use the abbreviations for the drugs as they previously mentioned in the "Introduction" section.

- Line no. 53, change "the output of Future Pharmaceutical Industries" into "the product of Future Pharmaceutical Industries".

-Page 4, line no. 8, change "The surfactant" into "The surfactants".

- Section 2.3:

- Please revise the significant figures in this section and all over the manuscript

- Line no. 20, replace the word "performed" as it is inappropriate.

-Line no. 23, remove the word " Hence".

- Line no. 26, change "where ETO was 10µg/mL, MOX was 4 µg/mL and NAL was 50 µg/mL". rewrite this sentence.

- Section 2.4.1:

- Page 5, lines no. 1- 21: under the title "Analysis of MOX, ETO, and NAL in their pharmaceutical formulation:" and page 8, lines 6 - 23, the authors described the analysis of separate dosage forms but not simultaneous determination. This is not matched with the title of the manuscript.

- Line no. 37, replace the sentence "was to the extent of" as it is inappropriate

-Lines no. 41-43, "Two eco-friendly synchronous fluorimetric processes (conventional and first derivative) were applied to estimate these drugs", there is no need for this sentence.

- Line no. 46, "where 1D", the word "where" here is not appropriate, replace it with a suitable word.

- Section 2.4.2:

- Line no. 3, "Moxavidex® 400 mg tablets for MOX were accurately weighed and then utterly crushed". The authors didn't clarify the number of tablets included in this study.

- Line no. 6, "Therefore, sonication", therefore here is not appropriate please replace it.

The authors used "Hence", "Therefore", "consequently" many times incorrectly at the beginning of the sentences, please, revise this through the manuscript.

- Line no. 8, "sonication for 20 min", please, explain why 20 min sonication was necessary. I think it is too much time for sonication.

- Line no. 10, "was implemented", replace this verb as it is not appropriate.

- Lines no. 11-22, rewrite this paragraph as it contains many mistakes and unclear descriptions.

- Lines no. 11-18, "While for ETO and NAL, an amount equivalent to 10 mg ETO of Etoposide Mylan® solution and 10 mg NAL of Nalufine® ampoule were individually moved into 100 mL volumetric flasks and completing to the mark with methanol and ethanol, respectively. Additionally, the achievement of a working standard solution for both ETO and NAL was applied by adding ethanol to the final mark". There is a contradiction in this paragraph. Please, clarify how you prepared the working solutions for ETO and NAL.

- Section 2.4.3:

- The authors described the analysis of each drug separately in spiked human urine but not simultaneous determination. This is not matched with the title of the manuscript. The authors must study the simultaneous determination of different mixtures with different ratios of the studied drugs in spiked human urine, not each drug alone. This is the main objective of the study as claimed by the authors.

- Lines no. 25-27, "Synchronous assessment of MOX, ETO, and NAL in the biological fluid was attained by involving human urine to design a standard calibration curve", this sentence is confusing and should be modified.

- Line no. 32, "and 3 mL of acetonitrile into a set of covered tubes", was this volume optimized? I think it is too much with respect to the greenness of the method. Why did you use acetonitrile not ethanol especially it is the diluting solvent in your study?.

- What do you mean by "covered tubes"? You should also mention their volumes.

5. Results and discussion:

- Page 5, Lines no. 40-51, the authors referred only to the figures and didn't explain in detail the synchronous spectrofluorimetric determination of the studied drugs.
- Where is the figure for the synchronous spectrofluorimetric determination of MOX?.
- Figures 2 and 3, MOX (0.08 μ g/mL) in Figure 2 gave fluorescence intensity of about 900, while in Fig. 3 when its concentration increased to 0.2 μ g/mL, the synchronous fluorescence intensity was about 400. It is well known that the synchronous spectrofluorimetry enhances the sensitivity not decreases it. Please, explain this contradiction.
- Figure 3, There is a big difference between the SF intensity when the drugs were determined simultaneously in a synthetic mixture and when each drug was determined separately especially for ETO and NAL. The two spectra (i.e in the synthetic mixture and separately) should give the same SF intensity.
- Remove "A" from the beginning of Figure 2 caption.
- Lines no. 42-44, "As illustrated in Fig. 02, MOX could be determined by the conventional spectrofluorimetric technique". You determined MOX using the synchronous spectrofluorimetric technique, not conventional spectrofluorimetry.
- Figures 4-6, please add the blank spectra to the figures.
- Page 6, line no. 3, "Experimental conditions impacting RFI", you didn't measure the RFI, rather you measured synchronous fluorescence intensity, please, correct this in section 3.1 and through the manuscript.
- Figure 7 caption, "Effect of different solvent", correct "solvent" into "solvents".
- Page 6, line no. 23, "Studying the pH effectiveness", replace "effectiveness" as it is inappropriate.
- Page 6, line no. 38, "1.0 % aqueous solution of SDS, CTAB, Tween-80, and β -CD macromolecules". Explain why the studied surfactants were used in this concentration (1.0 %)?.
- Figure 9 caption, "Effect of different surfactant", correct "surfactant" into "surfactants".
- Figures 7-9 can be collected in one figure.
- Section 3.1.4., please provide the figure for studying the effect of $\Delta \lambda$.
- Page 7, line no. 8, "applying ICH roles", change "roles" into "rules". The spelling is incorrect.
- Table 1, "Quantitation Limit of, QL (ng/mL)", remove "of".
- Section "3.2.3. Accuracy and precision", this section contains many mistakes and unclear descriptions. It should be revised and rewritten.
- The results of the proposed methods should be statically compared with the reported or reference methods to confirm the accuracy and precision in both pure forms and dosage forms. The authors compared them only in the dosage forms.

- Page 7, lines no. 28-30, "Accuracy of innovatory green synchronous spectrofluorimetric quantification of a ternary laboratory mixture of ETO, MOX, and NAL was applied". This sentence is confusing and should be modified.
- Page 7, line no. 30, "This performance was carried by", correct this sentence.
- Page 7, Lines no. 35-39, "Table 2 showed the accuracy of innovatory green synchronous spectrofluorimetric in conventional and derivative form for estimation of these coadministered drugs, and Fig. 09 confirmed this", How Figure 09 confirmed this?. Figure 09 is for the effect of different surfactants.
- Page 7, line no. 39, "Assessment precision of", correct this sentence.
- Page 7, line no. 42, "the consequence was abridged (Table 3), which approving excellent precision". This sentence is confusing and should be modified.
- Table 3, Change "% Found" in the heading of the table into "% Found \pm % RSD" and there is no need for the column of % RSD.
- Table 4, add the tabulated t and F values for Moxavidex tablets and Etoposide Mylan solution.
- Page 8, line no. 13, "and UV-detector was used for monitoring was at 254 nm". Revise this sentence.
- Page 8, section "3.3.1. Estimation of ETO, MOX, and NAL in pharmaceutical dosage forms". This section contains many mistakes and unclear descriptions; it should be revised well and modified.
- Page 8, line no. 30, "NAL is rapidly absorbed after intramuscular and is excreted unchanged". Revise this sentence.
- Page 8, line no. 31, "the SCF", this abbreviation was not defined before in the manuscript.
- Page 8, line no. 33, "Subsequently, the resulted were presented in (Fig. 10)", revise and correct this sentence.
- Page 8, lines no. 33-37, "A linear relationship by graphing (RFI) for MOX or the peak amplitude of the (1D) technique for ETO and NAL versus the final drug concentration ($\mu\text{g}/\text{mL}$) was approved". Rewrite this sentence.
- Page 8, lines no. 49-51, "Lastly, the monitoring of ETO, MOX, and NAL in human urine was involved in Table 5 and derivatized by utilizing the previous regression equations". This sentence is confusing and should be modified.
- Figure 10:
 - I think the caption is not related to the provided figure. The authors wrote "(a) Blank plasma", the study didn't use any plasma.
 - "(A) synchronous spectrofluorimetry (B) first derivative synchronous spectrofluorimetry for the determination of the studied drugs in spiked human urine:", Where is "A" and "B" in the provided figure?.

- The figure contains letters "a,b,c,d,e" and you defined a, b, and c only.
- The authors mentioned in the manuscript that they analyzed the studied drugs separately in the spiked human urine and provided the results in Table 5 but in the provided figure they claimed that they determined them as mixtures. There is contradiction between the figure and what explained in the manuscript and Table 5.
- Page 9, line no. 7, "based on concurred with identical criterion", revise and correct.
- Page 9, line no. 11, "Therefore, the results were significantly performed by mentioned criteria", It is better to change "significantly performed" with "assessed" or "evaluated".
- Page 9, line no. 21-24, "as GHS was categorized and characterized every chemical if the chemical penalty points are equal to two and one as a danger" or "warning", respectively." This sentence is confusing and should be corrected.
- Page 9, line no. 27, "Newly, Green Analytical Procedure Index (GAPI) has been established", revise and correct. "Newly" here is not appropriate.
- Page 9, line no. 30, "GAPI is recorded as the best tool", revise and correct.
- Page 9, line no. 37-41, "Furthermore, GAPI has been performed to the SF method involved in estimating the ETO, MOX, and NAL in human urine; it was recorded that yellow color for micro-extraction is applied red color (acetonitrile, a non-green solvent)". This sentence is confusing and should be modified.
- Table 6, in Analytical Eco-Scale, the calculation of penalty points for waste is incorrect. It should be 6, not 3. In this method, the waste was 10 mL = 3 penalty points and no treatment of the waste = 3 penalty points, so the total penalty points for waste should be 6.
- Table 6, you should add the reference ranges of Analytical Eco-Scale to the table footnotes.

6. Conclusion:

- Page 10, lines no. 3-13, this paragraph is confusing and contains many mistakes; it should be revised and modified.

Author's Response to Decision Letter for (RSOS-210683.R0)

See Appendix A.

RSOS-210683.R1 (Revision)

Review form: Reviewer 1

Is the manuscript scientifically sound in its present form?

Yes

Are the interpretations and conclusions justified by the results?

Yes

Is the language acceptable?

Yes

Do you have any ethical concerns with this paper?

No

Have you any concerns about statistical analyses in this paper?

No

Recommendation?

Accept as is

Comments to the Author(s)

All corrections have been made, the manuscript could be accepted as is

Decision letter (RSOS-210683.R1)

Dear Dr Salim:

Title: Eco-friendly fluorimetric approaches for the simultaneous estimation of the co-administered ternary mixture: etoposide, moxifloxacin, and nalbuphine
Manuscript ID: RSOS-210683.R1

It is a pleasure to accept your manuscript in its current form for publication in Royal Society Open Science. The chemistry content of Royal Society Open Science is published in collaboration with the Royal Society of Chemistry.

===COVID-SPECIFIC TEXT -- WILL ONLY BE ADDED TO COVID-PAPERS BY THE EDITORIAL OFFICE===

COVID-19 rapid publication process:

We are taking steps to expedite the publication of research relevant to the pandemic. If you wish, you can opt to have your paper published as soon as it is ready, rather than waiting for it to be published the scheduled Wednesday.

This means your paper will not be included in the weekly media round-up which the Society sends to journalists ahead of publication. However, it will still appear in the COVID-19 Publishing Collection which journalists will be directed to each week (<https://royalsocietypublishing.org/topic/special-collections/novel-coronavirus-outbreak>).

If you wish to have your paper considered for immediate publication, or to discuss further, please notify openscience_proofs@royalsociety.org and press@royalsociety.org when you respond to this email.

===END OF COVID-SPECIFIC TEXT -- WILL BE REMOVED AS NECESSARY BY THE EDITORIAL OFFICE===

Yours sincerely,
Dr Ellis Wilde
Publishing Editor, Journals

RSC Associate Editor
Comments to the Author:
(There are no comments.)

RSC Subject Editor
Comments to the Author:
(There are no comments.)

Reviewer(s)' Comments to Author:
Reviewer: 1

Comments to the Author(s)
All corrections have been made, the manuscript could be accepted as is

Appendix A

Response to Referees

Title: An eco-friendly fluorimetric approaches for the simultaneous estimation of antineoplastic, fluoroquinolone, and opioid analgesic drugs in human urine
Manuscript ID: RSOS-210683

Dear Editor-in-Chief,

On behalf of co-authors, I would like to thank the editorial board for giving us this chance to further revise our manuscript. We appreciate all the valuable comments raised by the reviewers. All the comments have been considered in preparing the revised version of the manuscript and a point-by-point reply is represented below:

Reviewer 1:

1. The article needs careful revision and editing.

The manuscript was revised carefully and corrected accordingly.

2. The title should be changed to be more representative.

The manuscript title was changed to be “Eco-friendly fluorimetric approaches for the simultaneous estimation of the co-administered ternary mixture: etoposide, moxifloxacin, and nalbuphine.”

3. The sensitivity of the Agilent device was sited at 800 v..... The mode of operation should be written instead.

The high voltage mode was added beside the value for more clarification and to ensure the method's reproducibility.

4. Under the procedure section, the Statement "completing the volume" must be changed to be " completing the mark."

The sentence "completed to the mark" was added accordingly.

5. In page# 4.....”Subsequently, these solutions were preserved in the fridge until utilized” ... at which degree?

The temperature degree (2 °C) was added to the sentence.

6. In procedure section: it is better to use the word mix instead of blend. Blending usually when the biological tissue is incorporated in the method.

The word blend was replaced by the suggested word “mix” accordingly. Thanks.

7. In Result and discussion section: why the authors used to number the figures in the zero non decimal way? please use the ordinary numbering formate.

Done.

8. In experimental optimization section:

- The abbreviation of Surface Active Agent (SAG) should be changed to SAA.

Done.

- Also "Therefore, these surfactants were not involved in this innovatory green process.....so what is the sence of trying this effect?"

Different surfactants and macromolecules were tested, seeking the high sensitivity of the proposed approaches [1, 2]. However, unfortunately, it does not work in our case [3, 4].

9. The dosage forms nominal content should be calculated, and the statistical obtained data should be added in Table #4.

The nominal content was calculated statistically, and results were abridged in table no. 4.

10. Figure #1: the image resolution should be improved....

The fig. 1 resolution was improved.

11. Figure #2 Please, point out the excitation and emission wavelengths used.

The excitation and emission wavelengths were added to the Figure.

12. Figures#4,5,6,8,9 should be transferred to the supplementary material data file.

The figures were moved to the supplementary material.

13. The intersection and symbols in figures 6 and 10 was displaced so need to be corrected.

Done

Reviewer:2

General comments:

1. This paper was not well written or revised, very poor English, many grammatical and spelling mistakes, and unclear descriptions. This paper can't be accepted in this form and English editing by a professional or a native English speaker is a must.

The manuscript was revised carefully, and the English language was edited and corrected accordingly.

2. The number of figures should be reduced and the resolution of the figures should be improved (at least 300 dpi).

According to reviewer 1 recommendations, 5 Figures were moved to the supplementary material. so, the number of figures were reduced to be 5 and the resolution was improved accordingly (300 dpi).

3. Please revise the significant figures in tables and all over the manuscript.

The significant figures were revised accordingly.

4. Some references don't follow the journal style and the journal names need to be abbreviated. For example, Ref. no. 11, 12, 13, 16, 18, 23, 24, 26, 30, 32, 33, 35.

All the references were revised and abbreviated accordingly.

5. Please revise the adjustment of the text within the manuscript to be uniform including the references.

Done

6. The statement "innovatory green processes" was unnecessarily repeated many times throughout the manuscript. Can you explain why?

The unnecessary repetition was removed from the manuscript and replaced with the most suitable scientific term.

Specific comments:

1. Title: The title of any paper is an important indication of its quality, so it should be correct and well revised.

The title was changed and revised accordingly.

- Please, remove "An" from the title as it is incorrect where "approaches" is a plural, not a singular word.

Done, thanks.

- Please, replace drug classes in the title with the name of actually studied drugs.

Done.

- In the title, the authors mentioned that the method was applied to human urine.

However, only spiked urine appeared in the manuscript and no real urine was analyzed. Analysis of spiked urine is not a guarantee for the successful application in real samples. So the authors should try to analyze real urine, otherwise, the urine

should be removed from the title.

The title was changed and the application in spiked human urine was removed from the title and discussed in the manuscript.

2. Abstract: It contains many mistakes and should be rewritten and revised well.

Done.

For example:

- Line no. 12, Change "leukemia cancer" into "leukemia" in the abstract and through the manuscript.

Done.

- Line no. 19 "Consequently, innovatory green processes for synchronous spectrofluorimetric quantification of a ternary mixture of antineoplastic, fluoroquinolone, and opioid analgesic drugs", Where is the sentence?. Please, write a complete sentence. There is no sentence without a verb.

Done.

- Line no. 25, Change "Conventional synchronous fluorimetric of MOX" into "Conventional synchronous fluorimetric determination of MOX".

Done.

- Line no. 27, change "first derivative synchronous fluorimetric of ETO and NAL" into " first derivative synchronous fluorimetric determination of ETO and NAL".

Done.

- Please, change the word "processes" into "methods" in the abstract and through the manuscript.

Done.

- Line no. 33, "The outcomes of these designed environmentally friendly processes, a linear correlation was described in the ranges of 0.04-0.40, 0.1-1.0, and 0.5-5.0 µg/mL for MOX, ETO, and NAL, respectively. This sentence is confusing and should be modified.

Done.

- Line no. 36, "Accordingly, several pharmaceutical dosages were achieved within the range simply and precisely". What do you mean by "achieved"?. Rewrite this sentence.

Done.

- Line no. 40, please replace "biological fluid" with "spiked human urine samples" through the manuscript and add some details in the abstract about the main results for the determination of the studied drugs in spiked human urine.

Done.

- Line no. 43, change "roles" into "rules". Its spelling is incorrect. This is a repeated mistake, correct it through the manuscript.

Done.

- Line no. 45, the abstract should not contain abbreviations such as NEMI and GAPI.

Done.

- Line no. 48, Keywords, change "Synchronous fluorimetric" into "Synchronous fluorimetry".

Done.

3. Introduction:

- Advantage of the proposed methods over the published ones should be introduced in the introduction.

Done.

The Introduction contains many writing mistakes:

- Line no. 6, "and is categorized as alkaloid 9-[4, 6-O-I-ethylidene-β-D-glucopyranoside]; (5S, 5aR, 8aS,9R)-9-(4,6-O-Ethylidene- β -D-glucopyranosyloxy)-5, 8, 8a, 9-tetra hydro -5-(4-hydroxy-3,5-dimethoxy phenyl)-isobenzofuro[5,6 f] [1,3] benzodioxol-6(5aH)-one [1-3]", This sentence is unclear, rewrite it.

Done.

- Line no. 13, change "MOX is named" into ""MOX is named as".

Done.

- Line no. 21, "USP pharmacopeia", correct it into "USP" or "United States pharmacopeia".

Done.

- Lines no. 23-38, rewrite this paragraph and make it simpler.

Done.

- Lines 39-42, "Hence, innovatory green processes for synchronous spectrofluorimetric (SF) quantification of a ternary mixture of ETO, MOX, and NAL in their pharmaceutical dosage form and/or biological fluids". Where is the sentence?. Please, write a complete sentence. There is no sentence without a verb.

Done.

- Line no. 45, "Since the selectivity of spectrofluorimetry is improved by applying synchronous scanning [39]", where is the rest of the sentence?.

Done.

-Line no. 47, change " the significance of" into "The significance of".

Done.

-Line no. 49, what do you mean by "is a manifestation of selectivity and sensitivity". Please rewrite it.

Done.

- Page 3, line no. 1, add "an" before "additional".

Done.

- Pages 3, line no. 2, "NEMI, GAPI", define the two abbreviations at their first appearance in the manuscript.

Done.

-Page 3, line no. 11, "The scope", I think this word is inappropriate, "The aim", "The objective", or "The goal" may be more appropriate.

Done.

- Page 3, line no. 19, change "roles" into "rules". The spelling is incorrect.

Done.

4. Experimental:

- Section 2.1:

- Please, add all details about the manufacturer of instruments i.e. the city and country....etc.

Done.

- The used cell should be written.

Done.

- Line no. 25, change "Fluorescence" into "synchronous fluorescence spectra".

Done.

- Line no. 28-30, "Thermostatically controlling and shaking for all experimental features was carried out by utilizing the England water bath of Cambridge Ltd". You didn't study the effect of temperature, what do you mean by this sentence?.

Explaining the type of the water bath used and the sentence was rewritten in an easier way.

-Line no. 33, "and separation using centrifuge model 2-16P (Germany)", correct this sentence.

Done.

- Section 2.2:

-Lines no. 39-44, use the abbreviations for the drugs as they previously mentioned in the "Introduction" section.

Done.

- Line no. 53, change "the output of Future Pharmaceutical Industries" into "the product of Future Pharmaceutical Industries".

Done.

-Page 4, line no. 8, change “The surfactant” into “The surfactants”.

Done.

- Section 2.3:

- Please revise the significant figures in this section and all over the manuscript

Done.

- Line no. 20, replace the word “performed” as it is inappropriate.

Done.

-Line no. 23, remove the word “ Hence”.

Done.

- Line no. 26, change “where ETO was 10µg/mL, MOX was 4 µg/mL and NAL was 50 µg/mL”. rewrite this sentence.

Done.

- Section 2.4.1:

- Page 5, lines no. 1- 21: under the title “Analysis of MOX, ETO, and NAL in their pharmaceutical formulation:” and page 8, lines 6 – 23, the authors described the analysis of separate dosage forms but not simultaneous determination. This is not matched with the title of the manuscript.

Actually, the main target of this work is to determine the co-administered drugs in the spiked human urine and additionally in their individual dosage forms as there is no co-formulated ones available right now.

- Line no. 37, replace the sentence “was to the extent of” as it is inappropriate

Done.

-Lines no. 41-43, “Two eco-friendly synchronous fluorimetric processes (conventional and first derivative) were applied to estimate these drugs”, there is no need for this sentence.

This sentence was deleted.

- Line no. 46, “where 1D”, the word “where” here is not appropriate, replace it with a suitable word.

Done.

- Section 2.4.2:

- Line no. 3, “Moxavidex® 400 mg tablets for MOX were accurately weighed and then utterly crushed”. The authors didn’t clarify the number of tablets included in this study.

The number of used tablets were added.

- Line no. 6, “Therefore, sonication”, therefore here is not appropriate please replace it.

Done.

The authors used "Hence", "Therefore", "consequently" many times incorrectly at the beginning of the sentences, please, revise this through the manuscript.

Done.

- Line no. 8, "sonication for 20 min", please, explain why 20 min sonication was necessary. I think it is too much time for sonication.

Sonication for 20 min is necessary to ensure the complete dissolution of the tables.

- Line no. 10, "was implemented", replace this verb as it is not appropriate.

Done.

- Lines no. 11-22, rewrite this paragraph as it contains many mistakes and unclear descriptions.

Done.

- Lines no. 11-18, "While for ETO and NAL, an amount equivalent to 10 mg ETO of Etoposide Mylan® solution and 10 mg NAL of Nalufine® ampoule were individually moved into 100 mL volumetric flasks and completing to the mark with methanol and ethanol, respectively. Additionally, the achievement of a working standard solution for both ETO and NAL was applied by adding ethanol to the final mark". There is a contradiction in this paragraph. Please, clarify how you prepared the working solutions for ETO and NAL.

Done.

- Section 2.4.3:

- The authors described the analysis of each drug separately in spiked human urine but not simultaneous determination. This is not matched with the title of the manuscript. The authors must study the simultaneous determination of different mixtures with different ratios of the studied drugs in spiked human urine, not each drug alone. This is the main objective of the study as claimed by the authors.

This section was rewritten in a clear way to show the procedure of the simultaneous determination of the studied drugs in spiked human urine.

- Lines no. 25-27, "Synchronous assessment of MOX, ETO, and NAL in the biological fluid was attained by involving human urine to design a standard calibration curve", this sentence is confusing and should be modified.

Done.

- Line no. 32, "and 3 mL of acetonitrile into a set of covered tubes", was this volume optimized? I think it is too much with respect to the greenness of the method. Why did you use acetonitrile not ethanol especially it is the diluting solvent in your study?.

Ethanol was tested firstly for protein precipitation, unfortunately it does not work.

So, the method selectivity was moved to use acetonitrile for complete protein precipitation and acceptable results were obtained.

- What do you mean by "covered tubes"? You should also mention their volumes.

The “covered tubes” term was modified to the more definite one and their volumes were added.

5. Results and discussion:

- Page 5, Lines no. 40-51, the authors referred only to the figures and didn't explain in detail the synchronous spectrofluorimetric determination of the studied drugs.

The synchronous spectrofluorimetric determination of the studied drugs was explained.

- Where is the figure for the synchronous spectrofluorimetric determination of MOX?.

- The figure for the synchronous spectrofluorimetric determination of MOX is represented in Figure 3 (ligand:3). It is marked by green color and superimposed with the synchronous fluorescence of the mixture (Figure 3(ligand:4)) which confirms the determination of MOX applying direct synchronous fluorimetry.

- Figures 2 and 3, MOX (0.08 $\mu\text{g/mL}$) in Figure 2 gave fluorescence intensity of about 900, while in Fig. 3 when its concentration increased to 0.2 $\mu\text{g/mL}$, the synchronous fluorescence intensity was about 400. It is well known that the synchronous spectrofluorimetry enhances the sensitivity not decreases it. Please, explain this contradiction.

Definitely, increasing or decreasing the sensitivity of the synchronous fluorimetry comparable to native fluorescence depends on the used $\Delta \lambda$. In native fluorescence, the excitation and emission fluorescence spectra of MOX was recorded at 292/460 nm. If the measurement was performed at $\Delta \lambda$ equal to the difference between the excitation and emission wavelength (about 168 nm), the sensitivity will be approximately the same in the native and synchronous fluorimetry. But unfortunately, this $\Delta \lambda$ wasn't able to separate the three drugs in mixture, so different $\Delta \lambda$ were tried to give good resolution of the studied drugs with adequate sensitivity. This is usually the main target of synchronous fluorimetry.

- Figure 3, There is a big difference between the SF intensity when the drugs were determined simultaneously in a synthetic mixture and when each drug was determined separately especially for ETO and NAL. The two spectra (i.e in the synthetic mixture and separately) should give the same SF intensity.

That is true; there is a difference between the SF intensity when the drugs were determined simultaneously in a synthetic mixture and when each drug was determined separately. This was monitored for ETO and NAL. As each drug has fluorescence readings at the wavelengths for the other, so when both drugs are determined as synthetic mixture there will be an additive effect due to contribution of fluorescence of each drug at the wavelength of the other. So, ETO and NAL couldnot be determined by direct synchronous fluorimetry and so, we

shifted to derivative fluorimetry for their separation. In contrast, MOX could be determined by direct synchronous fluorimetry as there is no difference between the drug alone and the mixture where they are superimposed on each other.

- Remove "A" from the beginning of Figure 2 caption.

Done.

- Lines no. 42-44, "As illustrated in Fig. 02, MOX could be determined by the conventional spectrofluorimetric technique". You determined MOX using the synchronous spectrofluorimetric technique, not conventional spectrofluorimetry.

Corrected.

- Figures 4-6, please add the blank spectra to the figures.

Blank spectra were added and shifted to supplementary data (figure S1-S3) as recommended by reviewer 1.

- Page 6, line no. 3, "Experimental conditions impacting RFI", you didn't measure the RFI, rather you measured synchronous fluorescence intensity, please, correct this in section 3.1 and through the manuscript.

Done.

- Figure 7 caption, "Effect of different solvent", correct "solvent" into "solvents".

Done.

- Page 6, line no. 23, "Studying the pH effectiveness", replace "effectiveness" as it is inappropriate.

Done.

- Page 6, line no. 38, "1.0 % aqueous solution of SDS, CTAB, Tween-80, and β -CD macromolecules". Explain why the studied surfactants were used in this concentration (1.0 %)?

1.0 % of SAA was used to form micelles.

- Figure 9 caption, "Effect of different surfactant", correct "surfactant" into "surfactants".

Done.

- Figures 7-9 can be collected in one figure.

According to reviewer 1 recommendations, the figures were moved to the supplementary material.

- Section 3.1.4., please provide the figure for studying the effect of $\Delta \lambda$.

Figure S6 was added showing the effect of $\Delta \lambda$.

- Page 7, line no. 8, "applying ICH roles", change "roles" into "rules". The spelling is

incorrect.

Done.

- Table 1, "Quantitation Limit of, QL (ng/mL)", remove "of".

Done.

- Section "3.2.3. Accuracy and precision", this section contains many mistakes and unclear descriptions. It should be revised and rewritten.

This section was revised and rewritten accordingly.

- The results of the proposed methods should be statically compared with the reported or reference methods to confirm the accuracy and precision in both pure forms and dosage forms. The authors compared them only in the dosage forms.

Done and the data obtained were abridged in Table 2.

- Page 7, lines no. 28-30, "Accuracy of innovatory green synchronous spectrofluorimetric quantification of a ternary laboratory mixture of ETO, MOX, and NAL was applied". This sentence is confusing and should be modified.

Done.

- Page 7, line no. 30, "This performance was carried by", correct this sentence.

Done.

- Page 7, Lines no. 35-39, "Table 2 showed the accuracy of innovatory green synchronous spectrofluorimetric in conventional and derivative form for estimation of these coadministered drugs, and Fig. 09 confirmed this", How Figure 09 confirmed this?. Figure 09 is for the effect of different surfactants.

Corrected.

- Page 7, line no. 39, "Assessment precision of", correct this sentence.

Done.

- Page 7, line no. 42, "the consequence was abridged (Table 3), which approving excellent precision". This sentence is confusing and should be modified.

Done.

- Table 3, Change "% Found" in the heading of the table into "% Found \pm % RSD" and there is no need for the column of % RSD.

Done.

- Table 4, add the tabulated t and F values for Moxavidex tablets and Etoposide Mylan solution.

Done.

- Page 8, line no. 13, "and UV-detector was used for monitoring was at 254 nm".
Revise this sentence.

Done.

- Page 8, section "3.3.1. Estimation of ETO, MOX, and NAL in pharmaceutical dosage forms". This section contains many mistakes and unclear descriptions; it should be revised well and modified.

Done.

- Page 8, line no. 30, "NAL is rapidly absorbed after intramuscular and is excreted unchanged". Revise this sentence.

Done.

- Page 8, line no. 31, "the SCF", this abbreviation was not defined before in the manuscript.

Corrected.

- Page 8, line no. 33, "Subsequently, the resulted were presented in (Fig. 10)", revise and correct this sentence.

Done.

- Page 8, lines no. 33-37, "A linear relationship by graphing (RFI) for MOX or the peak amplitude of the (1D) technique for ETO and NAL versus the final drug concentration ($\mu\text{g/mL}$) was approved". Rewrite this sentence.

Done.

- Page 8, lines no. 49-51, "Lastly, the monitoring of ETO, MOX, and NAL in human urine was involved in Table 5 and derivatized by utilizing the previous regression equations". This sentence is confusing and should be modified.

Done.

- Figure 10:

- I think the caption is not related to the provided figure. The authors wrote "(a) Blank plasma", the study didn't use any plasma.

Corrected.

- "(A) synchronous spectrofluorimetry (B) first derivative synchronous spectrofluorimetry for the determination of the studied drugs in spiked human urine:", Where is "A" and "B" in the provided figure?.

Added.

- The figure contains letters "a,b,c,d,e" and you defined a, b, and c only.

Added.

- The authors mentioned in the manuscript that they analyzed the studied drugs separately in the spiked human urine and provided the results in Table 5 but in the provided figure they claimed that they determined them as mixtures. There is contradiction between the figure and what explained in the manuscript and Table 5.

Sections 2.4.3. and 3.3.2. were rewritten in a clear way to show the procedure and the results of the simultaneous determination of the studied drugs in spiked human urine. Moreover, Table 5 was reorganized for the sake of simplicity.

- Page 9, line no. 7, "based on concurred with identical criterion", revise and correct.
Done.

- Page 9, line no. 11, "Therefore, the results were significantly performed by mentioned criteria", It is better to change "significantly performed" with "assessed" or "evaluated".
Done.

- Page 9, line no. 21-24, "as GHS was categorized and characterized every chemical if the chemical penalty points are equal to two and one as a danger" or "warning", respectively." This sentence is confusing and should be corrected.
Done.

- Page 9, line no. 27, "Newly, Green Analytical Procedure Index (GAPI) has been established", revise and correct. "Newly" here is not appropriate.
Done.

-Page 9, line no. 30, "GAPI is recorded as the best tool", revise and correct.
Done.

- Page 9, line no. 37-41, "Furthermore, GAPI has been performed to the SF method involved in estimating the ETO, MOX, and NAL in human urine; it was recorded that yellow color for micro-extraction is applied red color (acetonitrile, a non-green solvent)". This sentence is confusing and should be modified.
Done.

- Table 6, in Analytical Eco-Scale, the calculation of penalty points for waste is incorrect. It should be 6, not 3. In this method, the waste was 10 mL = 3 penalty points and no treatment of the waste = 3 penalty points, so the total penalty points for waste should be 6.
Done.

- Table 6, you should add the reference ranges of Analytical Eco-Scale to the table footnotes.
Done.

6. Conclusion:
- Page 10, lines no. 3-13, this paragraph is confusing and contains many mistakes; it should be revised and modified.
Done.

References

[1] F.A. Ibrahim, H. Elmansi, M.I. El-Awady, S. AboEl Abass, Investigation of micellar enhancement in simultaneous assay of rosuvastatin and amlodipine in their fixed-dose combined tablets, Microchemical journal 158(2020), 105207

[2] N.N. Atia, S.M. El-Gizawy, N.M. Hosny Facile micelle-enhanced spectrofluorimetric method for picogram level determination of febuxostat; application in tablets and in real human plasma, Microchemical journal, **147(2019)296-302**

[3] M.E. El Sharkasy, M. Walash, F. Belal, M.M. Salim, Conventional and first derivative synchronous spectrofluorimetric methods for the simultaneous determination of cisatracurium and nalbuphine in biological fluids, *Spectrochim Acta A Mol Biomol Spectrosc*, **228 (2020) 117841**.

[4] R.I. El-Bagary, E.F. Elkady, N.A. Farid, N.F. Youssef, Validated spectrofluorimetric methods for the determination of apixaban and tirofiban hydrochloride in pharmaceutical formulations, *Spectrochim Acta A Mol Biomol Spectrosc*, **174 (2017) 326-330**.

Thanks in Advance.

Regards,

Mohamed M. Salim, Ph.D.